# Multidisciplinary Unit Improves Pregnancy Outcomes in Women with Rheumatic Diseases and Hereditary Thrombophilias: An Observational Study

**DOI:** 10.3390/jcm10071487

**Published:** 2021-04-03

**Authors:** Isabel Añón-Oñate, Rafael Cáliz-Cáliz, Carmen Rosa-Garrido, María José Pérez-Galán, Susana Quirosa-Flores, Pedro L. Pancorbo-Hidalgo

**Affiliations:** 1Department of Rheumatology, Jaén University Hospital, 23007 Jaén, Spain; mjpg77@hotmail.com; 2Department of Rheumatology, Virgen de las Nieves University Hospital, 18012 Granada, Spain; rcalizcaliz@gmail.com (R.C.-C.); susanaq_06@hotmail.com (S.Q.-F.); 3Department of Statistics, Foundation FIBAO, Jaén University Hospital, 23007 Jaén, Spain; crosa@fibao.es; 4Department of Nursing, Faculty of Health Sciences, University of Jaén, 23003 Jaén, Spain; pancorbo@ujaen.es

**Keywords:** rheumatic diseases, systemic lupus erythematosus, hereditary thrombophilias, antiphospholipid syndrome, pregnancy, miscarriage, foetal death, multidisciplinary consultation

## Abstract

Rheumatic diseases (RD) and hereditary thrombophilias (HT) can be associated with high-risk pregnancies. This study describes obstetric outcomes after receiving medical care at a multidisciplinary consultation (MC) and compares adverse neonatal outcomes (ANOs) before and after medical care at an MC. This study is a retrospective observational study among pregnant women with RD and HT treated at an MC of a university hospital (southern Spain) from 2012 to 2018. Absolute risk reduction (ARR) and number needed to treat (NNT) were calculated. A total of 198 pregnancies were registered in 143 women (112 with RD, 31 with HT), with 191 (96.5%) pregnancies without ANOs and seven (3.5%) pregnancies with some ANOs (five miscarriages and two foetal deaths). Results previous to the MC showed 60.8% of women had more than one miscarriage, with 4.2% experiencing foetal death. MC reduced the ANO rate by AAR = 60.1% (95%CI: 51.6−68.7%). The NNT to avoid one miscarriage was 1.74 (95%CI: 1.5–2.1) and to avoid one foetal death NNT = 35.75 (95CI%: 15.2–90.9). A total of 84.8% of newborns and 93.2% of women did not experience any complication. As a conclusion, the follow-up of RD or HT pregnant women in the MC drastically reduced the risk of ANOs in this population with a previous high risk.

## 1. Introduction

Rheumatic diseases (RD) can affect women of reproductive age [1]. These rheumatic diseases are systemic lupus erythematosus (SLE), antiphospholipid syndrome (APS), Sjögren syndrome (SS), rheumatoid arthritis (RA), spondyloarthritis (SpA), and undifferentiated connective tissue disease (UCTD) [2]. Due to the coexistence of RD and pregnancy, there is need for a collaborative approach between various specialties such as rheumatology, internal medicine, haematology, obstetrics, and neonatology. In addition, the search for prothrombotic factors causing recurrent foetal loss increases the number of cases of hereditary thrombophilias (HT) during reproductive age. 

During pregnancy, there are hormonal and immunological changes which keep maternal–foetal immune tolerance [3]. As pregnancy develops, the cellular component decreases, and the humoral component increases [4,5]. Pregnancies develop differently in accordance with the immunological alteration that triggers each RD (Table 1). RA alters cellular immunity, and there is a trend towards improvement during pregnancy [6,7]. Conversely, SLE alters humoral immunity increasing the risk of a flare, especially during the third term of pregnancy [7,8,9].

On the other hand, pregnant women have a physiological hypercoagulation state caused by an increase in clotting factors, decrease in anticoagulant proteins, and fibrinolysis [10,11]. This state becomes exacerbated in patients with thrombophilias [12,13].

RDs and HTs are associated with high-risk pregnancies. There is a higher frequency of adverse neonatal outcomes (ANOs) such as miscarriage and foetal death and maternal–foetal complications such as preeclampsia, prematurity, intrauterine growth retardation (IUGR), neonatal lupus, and congenital heart block (CHB) [14,15,16]. Maternal pathology and maternal antibody positivity, in particular for aPL, anti-Ro/SSA, and anti-La/SSB antibodies, are related to increased obstetric risk [17,18,19].

Monitoring pregnant women with RD or HT in multidisciplinary consultations (MC) allows for individualised follow-up and treatment protocols that improve prognosis [19,20]. Multidisciplinary teams make possible preconception counseling, obstetric risk stratification, evaluation of pharmacology safety during pregnancy, and control of maternal–foetal complications [21,22,23,24].

In conclusion, there is a need to take the effects of pregnancy into consideration regarding maternal pathology, the impact of disease activity on the foetus, and pharmacology safety during pregnancy [25,26]. MC is the best method to rigorously monitor pregnancy in women with RD and HT. This study aimed to describe the obstetric outcomes after receiving medical care at an MC and compare the occurrence of ANOs before (standard care) and after receiving medical care at an MC.

## 2. Methods

### 2.1. Design and Study Population

A retrospective observational study based on data obtained from clinical practice for pregnant women with RD and HT. Women treated at an MC for rheumatic diseases and pregnancy at a university hospital integrated into the Public Health Care System of Andalusia, (in Spanish, Servicio Andaluz de Salud), in the city of Granada (Spain) between 2012 (when the MC service started) and 2018. All pregnancy episodes of women during this period when followed by the MC were registered and compared with pregnancy episodes of these women in previous years, when followed by standard care.

The MC used a preferential care circuit between obstetrics, haematology, and rheumatology. Patients were referred to the rheumatic diseases and pregnancy unit for joint follow-up and close monitoring by these medical specialties. The waiting time for access was less than one week, and thereafter patients were examined every 4 to 6 weeks during the entire pregnancy by the obstetrics and rheumatology services.

In standard care, there was no organised collaboration between obstetrics, haematology and rheumatology. Each patient was followed independently by each speciality, without a specific preferential care circuit. In rheumatology, pregnant patients were examined with the same frequency as non-pregnant patients, every 4 to 6 months; thus, they were not closely monitored.

The inclusion criteria were pregnant women aged ≥ 18 years diagnosed with RD or HT, clinically stable over the previous six months, and treated with medicines that were approved as safe for pregnancy administration.

Data were excluded from pregnancy episodes not having complete follow-up at an MC and women with missing data on ANOs that occurred before follow-up.

Due to the study characteristics, an estimate of sample size was not performed prior to the study, and all women who were treated at the MC and fulfilled the selection criteria during the study period were consecutively recruited.

### 2.2. Data Collection

Data were obtained from each patient’s medical electronic record. Data on demographic variables, baseline test, and data obtained during MC follow-up were also collected.

#### 2.2.1. Baseline Variables

Before the first pregnancy monitored at an MC, the following data were collected: maternal age at first pregnancy, maternal pathology, type of HT, comorbidity associated with RD, positivity of aPL, anti-Ro/SSA, and anti-La/SSB. Positive aPL tests were considered when lupus anticoagulant (LA) was present in plasma on two or more occasions at least 12 weeks apart, detected according to the guidelines of the International Society on Thrombosis and Haemostasis. Anticardiolipin antibody (aCL) and anti-β2-glycoprotein-I antibody of IgG and/or IgM isotype were present in serum or plasma, on two or more occasions, at least 12 weeks apart, as measured by a standardised ELISA. Regarding aPL antibodies, double or triple positivity was considered if the woman had two or three positive aPL tests at the same time: lupus anticoagulant (LA), anticardiolipin antibody (IgG or IgM), or anti-β2-glycoprotein (IgG or IgM). All women in the cohort had received standard care prior to MC; thus, the following variables were recorded during standard care: ANOs, neonatal lupus history, CHB, and deep vein thrombosis (DVT). A miscarriage before week 12 of pregnancy and foetal death after week 12 of pregnancy not explained by chromosomopathies, malformations, or infections were considered to be ANOs.

#### 2.2.2. Follow-Up Variables

Each individual pregnancy followed at the MC was considered to be a pregnancy episode. During follow-up at the MC, if the patient had a twin pregnancy or several pregnancies, they were each treated as different pregnancy episodes. Maternal age, treatment received, and ANOs or successful neonatal outcomes were registered for each pregnancy episode. ANOs in standard care prior to MC were compared to those registered during follow-up at an MC.

As for successful pregnancy episodes, length of the pregnancy, type of childbirth (eutocic delivery, instrumental delivery, or Caesarean section), newborn weight and gender, foetal complications (prematurity, IUGR, neonatal lupus, and CHB) and maternal complications (DVT, preeclampsia, and flare of maternal pathology) were registered. Prematurity was considered when birth occurred before the 37th week of pregnancy, IUGR when the newborn’s weight was under the 10th percentile for that gestational age, neonatal lupus according to laboratory test results or skin lesions, and CHB if observed on foetal echocardiogram.

### 2.3. Data Analysis

Quantitative variables were expressed as the mean and standard deviation (SD), and nominal variables as frequency and percentage.

Differences among subjects’ characteristics, treatments received, and pregnancy episode profiles, in accordance with the pathology, were studied with the Chi-Square test or Fisher exact test, and the Kruskal–Wallis test was used for multiple comparisons. The Shapiro-Wilk test was used to assess numeric variable normality. The McNemar and Wilcoxon tests were used to compare ANOs that occurred before and during MC. In order to assess the effect size in the differences between ANOs that occurred before and during MC, the absolute risk reduction (ARR) and the needed number of women to be treated (NNT) with corresponding 95%CI were calculated. Data are displayed separately for the two types of pathologies, RD and HT.

For all analyses, a *p* value < 0.05 is considered statistically significant. Software used to carry out statistical analyses were Version 21 of IBM^®^ SPSS^®^ and R version 4.0.3.

### 2.4. Ethical Aspects

The study was conducted according to the Declaration of Helsinki, and the Research Ethics Committee of Granada approved the protocol. Since it used anonymised and retrospective data, informed consent was not requested with the approval of the Ethics Committee.

## 3. Results

### 3.1. Patient Characteristics

The final sample included 198 pregnancy episodes in 143 women diagnosed with RD (*n* = 112) or HT (*n* = 31).

Mean maternal age at first pregnancy was 33.6 (SD 5.4) years, and 44.8% of patients were ≥ 35 years old. The distribution according the specific type of RD was SLE in 49 (34.3%) women, primary APS in 40 (28%), RA in six (4.2%) women, and other RDs in 17 (11.8%) (nine women with SS, seven with UCTD, and one with SpA). Moreover, there were 31 women (21.7%) with a diagnostic of primary HT. Data disaggregated by pathologies are shown in Table 2. Higher comorbidity and younger age stand out in the SLE group. The 48 women with APS, primary and secondary to SLE, had the following aPL profile: 15 (31.3%) triple positivity, 11 (22.9%) double positivity, and 28 (58.3%) AL positive. Additionally, 33 women were anti-Ro/SSA positive, and 9 were anti-La/SSB positive. Out of these 42 women, 19 had SLE.

Out of the 55 women with some type of thrombophilia, 31 were primary HT, but 24 were secondary to a RD. As for types of HT: 25 (45.5%), alteration of the methylenetetrahydrofolate reductase (MTHFR) gene; 20 (36.4%), factor XII gen alteration; 5 (9.1%), alteration in prothrombin; 4 (7.3%), protein S deficiency; and 1 (1.7%), factor V Leiden alteration.

### 3.2. Pregnancy Outcomes

Table 3 shows the number of ANOs, thrombotic episodes, and CHB in women followed in standard care and those treated at an MC. Up to 87 (60.8%) women had more than one miscarriage episodes when treated in standard care (prior to the MC). Those groups with a higher percentage of miscarriages were the following: among women with RD, APS, 87.5%; SLE, 40.8%; other RDs, 35.3%; and Ras, 33.3%; and in women with primary HT, 77.4%. Within the primary APS group, 80% of women had triple positivity. Six women (4.2%) had experienced previous foetal death (three women in the primary HT, one in the SLE, one in the APS group, and one in the RD group).

As for foetal complications, one woman with SLE (0.7%) had a neonatal lupus case with CHB complication while followed in standard care. No cases occurred when treated at an MC.

There were also differences regarding thrombotic episodes. Up to 15 (10.5%) women had more than one DVT episode at standard care: five DVT episodes occurred in the primary HT group, four episodes in the SLE and APS group, and two in other RD. One case of DVT was registered in women treated at an MC.

Overall (RD and HT combined), the ARR for miscarriage was 57.3% (48.8–65.9), with NNT= 1.7 (1.5–2.1); for foetal death, the ARR was 2.8 (%1.1–6.6), with NNT= 35.6 (15.2–90.9) and for DVT, the ARR was 9.8% (4.6–14.9), with NNT= 10.2 (6.7–21.8).

#### 3.2.1. Outcomes in the MC Group

A total of 198 pregnancy episodes were registered in the group of 143 women who were followed at the MC. Out of these 198 episodes, 191 (96.5%) had a successful neonatal outcome, while seven (3.5%) had some ANOs: five miscarriages and two foetal deaths (Table 3 and Table 4).

Care provided at an MC reduced the rate of ANOs by ARR = 60.1% (95%CI: 51.6−68.7%). The miscarriage rate decreased by ARR = 57.3% (95%CI: 48.8–65.9), and the foetal death rate decreased by ARR = 2.8% (95%CI: 1.1–6.6). Additionally, a decrease in ARR = 66.7% (95%CI: 40.1–93.2) was observed in the group with the highest number of prior miscarriages (those with antiphospholipid syndrome and triple positivity). These data mean an NNT = 1.74 (95%CI: 1.5–2.1) of women should be treated to avoid one miscarriage, and an NNT = 35.75 (95%CI: 15.2–90.9) to avoid one foetal death. Furthermore, 47 (23.7%) pregnancy episodes developed in women who were anti-Ro/SSA or anti-La/SSB positive, and none of the newborns had neonatal lupus or CHB.

#### 3.2.2. Therapies Used in the MC Group

Table 5 shows the specific therapies received by the pregnant women followed at an MC. A total of 124 (62.6%) women were treated with low-molecular-weight heparin (LMWH); most of them (74.2%) received enoxaparin at doses of 40 mg a day.

A total of 111 (56.1%) women received low doses of acetyl salicylate acid (ASA); it is noteworthy that 91.2% of women in the primary APS group received it. In addition to the standard treatment with LMWH and ASA, a group of high-risk women (*n* = 16) with recurrent miscarriages received treatment with intravenous immunoglobulins (at doses of 0.4 mL/kg on day 0, 1, and every 3 weeks until the end of pregnancy).

Up to 85 (42.9%) women were treated with hydroxychloroquine (HCQ), most of them (91.8%) diagnosed with SLE. Therapies used with less frequency were anti-TNF treatment for three women (1.5%) until the second term of pregnancy (certolizumab for two women with RA and etanercept for one woman with SpA) and corticosteroids (usually doses of ≤ 5 mg of prednisone) for 30 (15.2%) women.

### 3.3. Neonatal Outcomes

#### 3.3.1. Neonatal Characteristics

A total of 191 newborns were analysed. Weeks of gestation were (mean) 36.9 (SD 7.4), and newborn weight was (mean) 2817.6 (SD 938.3) grams. Table 6 displays these data.

The analysis found statistically significant differences between the weeks of gestation of these groups: SLE and HT, *p* = 0.039, SLE and other RDs *p* = 0.003, and APS and other RDs *p* = 0.040. There were also statistically significant differences found in the newborns’ weight for these groups: SLE and HT, *p* = 0.008. These statistically significant differences have been proven to be due to the various studied pathologies and not due to a difference of maternal age at first pregnancy, as Table 2 shows.

#### 3.3.2. Deliveries

A total of 98 labours (51.3%) were eutocic, 68 (35.6%) had a C-section, and 25 (13.1%) instrumental delivery, mostly vacuum extraction (56%). There were no statistically significant differences between different types of delivery according to maternal pathology (*p* = 0.918). Most (91%) of the indications for C-section or instrumental delivery were obstetric, and only 7.5% were due to maternal pathology. The obstetric reasons were the following: 35.5% failed induction, 18.8% to provide support during expulsion stage, 17.6% loss of foetal wellbeing, 15.3% foetal malposition, 9.4% twin pregnancy, and 3.6% premature rupture of membranes. Maternal causes were preeclampsia and one patient who had previously undergone renal transplant.

#### 3.3.3. Neonatal Complications

A total of 84.8% of newborns did not have any neonatal complications. The neonatal complications identified were the following: 10 (5.2%) newborns had IUGR (mostly in women (*n* = 6) having primary APS); 22 newborns (11.5%) were born prematurely (in women with SLE or primary APS). No case of neonatal lupus or CHB was registered

#### 3.3.4. Maternal Complications

Overall, 93.2% of women did not have any maternal complications. In women with SLE, seven preeclampsia, one polyarticular flare, and one postpartum haemorrhage were registered. In women with primary APS, there were three cases of preeclampsia and one DVT. There were no maternal complications in the women with the others diseases.

## 4. Discussion

At the present time, the management of pregnancy in women with RD and/or with HT continues to be a challenge, although there has been an increase in successful pregnancy episodes. This is due to specialized management, in many cases by means of the creation of MCs composed of rheumatology, internal medicine, haematology, obstetrics, and neonatology units. [25,27]

This study reports on outcomes obtained from real clinical practice in a MC unit for pregnant women with RD or HT; therefore, there is a significant clinical diversity that greatly contributes to the study. Information from 198 pregnancy episodes was obtained from 143 women with different diagnoses, 112 with RD and 31 with HT. The number of pregnancy episodes registered for each pathology was similar to those reported in other studies [28,29].

In coordination with the haematology department, 55 women with some type of thrombophilia were treated: 31 were primary HT and 24 were secondary to a RD. These 24 women with RD and secondary thrombophilia had recurrent miscarriages of no apparent obstetric cause and aPL negativity was identified. Through follow-up of women at the MC, as many as 95.1% (39/41) of the pregnancy episodes of women with primary HT were successfully terminated, despite these women having suffered a mean of 2.7 (SD 2.2) previous miscarriages. The high risk of ANOs in women with HT has been pointed out by several studies; a meta-analysis went over the risk of foetal loss in HT and obtained an odds ratio (OR) that ranged between 2.7 (95%CI: 1.3–5.6) and 1.68 (95%CI: 1.1–2.6) depending on the type of HT [30].

Preconceptional clinical records of ANOs before being treated at the MC helped to stratify the risk, plan the pregnancy, and evaluate pharmacological safety. The main recommendation and one of the inclusion criteria was the clinical stability of the disease in the previous six months using safe medication. In this study, 198 pregnancy episodes were registered in the MC, and 191 (96.5%) of them had successful outcomes. These outcomes are comparable to similar clinical practice studies: 95.1% and 84.8% in a Turkish cohort and in a Japanese cohort, respectively [28,29].

The high frequency of ANOs (65%) recorded in these women when followed by standard care was substantially reduced when treated in the MC unit, both in women with RD and with HT. This can be considered as a great achievement, especially given that 44.8% of these women were ≥ 35 years and a lot of them (62.3%) were diagnosed with SLE or APS. Both SLE and age over 35 years area considered as major risk factors for foetal adverse events (OR 7.4; 95%CI: 1.3–40.8, *p*  = 0.021) [29]. In our study, the frequency of ANOs in standard care was 42.8% in SLE and 90% in APS. However, after the introduction of the MC, the ARR of ANOs was 36.7% (95%CI: 21.3–52.1) in the SLE group and 87.5% (95%CI: 77–98) in the APS group. Although the present study registered higher previous ANOs in the APS group, the ARR after MC was higher. This could be explained by the influence of the type of RD existing during pregnancy, because the SLE is the pathology with the highest risk to reactivate during pregnancy and with the greatest number of complications associated. This phenomenon of SLE reactivation during pregnancy, as observed by the PROMISSE study, is influenced by different risk factors, despite the control of disease activity before pregnancy [31]. In two different clinical practice studies with Iranian and Chinese women with SLE, a frequency of disease reactivation of 45% and 38% was registered during pregnancy, respectively [32,33]. Furthermore, risk of foetal loss with APS has been widely reported in literature [28,34]. A systematic revision with meta-analysis compared foetal loss in 556 pregnant women with SLE versus 385 women with APS; without the appropriate treatment, the group with APS presented a higher risk of foetal loss (RR: 4.49; 95%CI: 2.1–9.6; *p* = 0.0001) [35].

The type and percentage of maternal–foetal complications identified in this study were similar to those cited in the literature: prematurity (11.5%), IUGR (IUGR), and preeclampsia (5.2%), mostly in women with SLE or APS.

In women with APS, our results (16.1 prematurity, 10.7% IUGR, and 5.4% preeclampsia) are similar to those reported in a meta-analysis from 770 pregnant women with APS which found a higher risk of IUGR (RR = 1.4 (95%CI: 1.1–1.8; *p* = 0.02) and prematurity RR = 1.86 (95%CI: 1.5–2.3; *p* = 0.0001). [36] Another study with 58 pregnant women observed that aPL positivity was associated with a higher risk of preeclampsia (OR 2.2; 95%CI: 1.1–4.3; *p* = 0.016) [37].

In women with SLE, our study found a 10% of preeclampsia, which is consistent with some previous reported data from meta-analysis (preeclampsia in SLE RR = 1.9 (95%CI: 1.4–2.5; *p* = 0.00001) [38]. No case of CHB or neonatal lupus was found; this might be due to the protective effect HCQ, although further studies are required to confirm this. A study of 257 pregnancies with anti-Ro/SSA (40 under HCQ treatment and 217 without HCQ) associated the HCQ treatment with being a protective factor against CHB (OR 0.23; 95%CI: 0.06–0.92; *p* = 0.037) [39].

Some limitations of the present study that have to be taken into account are the retrospective design and lack of objective measures regarding clinical activity for each one of these pathologies. A selection bias is also possible because more than one pregnancy was registered for 31.5% of women, and 2.8% women had a twin pregnancy. There can also be confounding biases that have not been measured since morbidity during pregnancy depends on multiple factors, and not all these variables are controllable. The study reports data on women with two types of pathologies (rheumatic diseases and thrombophilias) that could be considered separately. However, both types have in common a high rate of pregnancy complications and adverse neonatal outcomes; thus, from an obstetric perspective, it might make sense to consider both types as high-risk pregnancies. Additionally, the aim was to report obstetric outcomes from a real-life multidisciplinary practice.

Despite being a single-centre study, it could have a significant impact in real clinical practice. The outcomes of the present study could be considered as preliminary data to design prospective studies with larger samples. Although a number of cohort studies with SLE and APS can be found in the literature, there are only a few that examine the effect of the multidisciplinary approach as proposed in MC.

## 5. Conclusions

This study highlights the high rate of successful neonatal outcomes after a multidisciplinary management of pregnant women with RD or HT, when provided by a team with rheumatologists, obstetricians, and haematologists. The management of these high-risk pregnancies in the MC achieved very satisfactory obstetric and neonatal outcomes, with 96.5% of live newborns and 84.8% of labours without foetal complications and 93.2% of women without maternal complications. These outcomes are even more significant considering that it was a population with a previously high percentage of ANOs before starting management in the MC unit. These outcomes could be useful to promote multidisciplinary care for pregnancy in women with RD or HT.

## Figures and Tables

**Table 1 jcm-10-01487-t001:** Rheumatic diseases and hereditary thrombophilia evolution during pregnancy.

Pathology	Evolution in Pregnancy	ANOs and Complications
SLE	Worsens	Miscarriage, foetal death, prematurity, IUGR, neonatal lupus, CHB, preeclampsia, and C-section.
RA	Improves in pregnancy, worsens in postpartum	Prematurity, IUGR, C-section, neonatal lupus, and CHB
SpA	Improves in pregnancy, worsens in postpartum	Prematurity, IUGR, and C-section
APS	Irrelevant, depends on aPL	Miscarriage, foetal death, prematurity, IUGR, preeclampsia, and C-section
SS	Irrelevant	Prematurity, IUGR, C-section, neonatal lupus, and CHB
UCTD	According to clinical status	Miscarriage, foetal death, prematurity, IUGR, neonatal lupus, CHB, preeclampsia, and C-section
HT	Irrelevant, depends on mutation	Miscarriage, foetal death, prematurity, IUGR, preeclampsia and C-section

Adverse neonatal outcomes (ANOs), rheumatic diseases (RD), hereditary thrombophilia (HT), systemic lupus erythematosus (SLE), intrauterine growth retarded (IUGR), Caesarean section (C-section), neonatal lupus, congenital heart block (CHB), rheumatoid arthritis (RA), spondyloarthritis (SpA), antiphospholipid syndrome (APS), antiphospholipid (aPL) antibodies, Sjögren syndrome (SS), undifferentiated connective tissue disease (UCTD). Taken from 6, 18.

**Table 2 jcm-10-01487-t002:** Baseline characteristics of women treated at an MC by type of disease.

	Rheumatic Diseases				Thrombophilias	
	SLE (*n* = 49)	Primary APS (*n* = 40)	Other RD (*n* = 17)	RA (*n* = 6)	Primary HT (*n* = 31)	*p* Value *
Frequency (%)						
Secondary SS	5 (10.2)	0 (0)	0 (0)	0 (0)	0 (0)	0.084
Secondary APS	8 (16.3)	0 (0)	0 (0)	0 (0)	0 (0)	0.005
Secondary Thrombophilia	10 (20.4)	10 (25)	3 (17.6)	1 (16.7)	-	0.019
Lupus nephritis	15 (30.6)	0 (0)	0 (0)	0 (0)	0 (0)	0.002
Age at first pregnancy, mean (SD)	31.5 (5.5)	34.7 (5.1)	33.70 (5.9)	36.37 (2.2)	34.9 (4.9)	0.014 **
Women older than 35	13 (26.5)	20 (50)	9 (52.9)	5 (83.3)	17 (54.8)	0.015

* Comparison of baseline characteristic by pathology (Fisher’s exact test, Kruskal–Wallis test, and Pearson’s Chi-Squared test) ** Multiple comparisons do not have enough statistical power to detect which groups there are differences between multidisciplinary consultation (MC), systemic lupus erythematosus (SLE), antiphospholipid syndrome (APS), hereditary thrombophilia (HT), rheumatic disease (RD), rheumatoid arthritis (RA), Sjögren syndrome (SS), and standard deviation (SD).

**Table 3 jcm-10-01487-t003:** Comparison of ANOs, thrombotic episodes, and CHB in 143 pregnant women treated in standard care and when treated at an MC.

	RD (*n* = 112)		*p* Value *	HT (*n* = 31)		*p* Value *
	Standard Care	MC		Standard Care	MC	
No miscarriage	49 (43.8%)	109 (97.3%)	<0.001	7 (22.6%)	29 (93.5%)	<0.001
One miscarriage	20 (17.9%)	3 (2.7%)		3 (9.7%)	2 (6.5%)	
Two miscarriages	21 (18.8%)	0 (0%)		6 (19.4%)	0 (0%)	
Three or more miscarriages	22 (19.6%)	0 (0%)		15 (48.4%)	0 (0%)	
Total miscarriages	149	3		82	2	
ARR (95% CI)	54 (44–63)			71 (54–88)		
NNT	1.87 (1.58–2.28)			1.41 (1.14–1.86)		
Miscarriages, mean (SD)	1.33 (1.6)	0.026 (0.2)	<0.001	2.64 (2.1)	0.64 (0.2)	<0.001
Foetal deaths	3 (2.7%)	2 (1.8%)	1.000	3 (9.7%)	0 (0%)	0.250
ARR (95% CI)	1 (0–5)			10 (0–20)		
NNT	112 (21–Inf.)			10.33 (4.98–Inf.)		
CHB	1 (0.9%)	0 (0%)	NA	0 (0%)	0 (0%)	NA
DVT	10 (8.9%)	0 (0%)	0.002	5 (16.1%)	1 (3.2%)	0.219
ARR (95% CI)	9 (4–14)			13 (0–27)		
NNT	11.20 (7.0–27.4)			7.7 (3.6–Inf.)		

* (Pearson’s Chi-Squared test, Wilcoxon test, McNemar’s test). Adverse neonatal outcomes (ANOs), multidisciplinary consultation (MC), absolute risk reduction (ARR), number needed to treat (NNT), confidence interval (CI), standard deviation (SD), congenital heart block (CHB), infinite (inf), calculation of statistical test not applicable because the table structure does not allow for it (NA), and deep vein thrombosis (DVT).

**Table 4 jcm-10-01487-t004:** Pregnancy episodes in women followed at the MC by pathology.

	Rheumatic Diseases				Thrombophilias	
	SLE (*n* = 73)	Primary APS (*n* = 57)	Other RD (*n* = 21)	RA (*n* = 6)	Primary HT (*n* = 41)	*p* Value
Frequency (%)						
One pregnancy	30 (61.2)	25 (62.5)	15 (88.2)	6 (100)	21 (67.7)	0.034 *
Two pregnancies	14 (28.6)	13 (32.5)	0 (0)	0 (0)	10 (32.3)	
Three pregnancies	5 (10.2)	2 (5)	2 (11.8)	0 (0)	0 (0)	
Neonatal outcomes						
Miscarriage follow-up	1 (1.4)	1 (1.8)	1 (4.8)	0 (0)	2 (4.9)	0.646 **
Foetal death follow-up	2 (2.7)	0 (0)	0 (0)	0 (0)	0 (0)	
Newborns	70 (95.9)	56 (98.2)	20 (95.2)	6 (100)	39 (95.1)	

* Comparison of number of pregnancies by pathology (Fisher’s exact test). ** Comparison of neonatal outcomes by pathology. Multidisciplinary consultation (CM), systemic lupus erythematosus (SLE), antiphospholipid syndrome (APS), hereditary thrombophilia (HT), rheumatic disease (RD), and rheumatoid arthritis (RA).

**Table 5 jcm-10-01487-t005:** Therapies during pregnancy episodes by pathology.

	Rheumatic Diseases				Thrombophilias	
	SLE (*n* = 73)	Primary APS (*n* = 57)	Other RD (*n* = 21)	RA (*n* = 6)	Primary HT (*n* = 41)	*p* Value *
Frequency (%)						
LMWH during pregnancy	26 (35.6)	55 (96.5)	5 (23.8)	1 (16.7)	37 (90.2)	<0.001
LMWH postpartum	38 (52.1)	57 (100)	9 (42.9)	1 (16.7)	40 (97.6)	<0.001
ASA	33 (45.2)	52 (91.2)	7 (33.3)	0 (0)	19 (46.3)	<0.001
HCQ	67 (91.8)	0 (0)	16 (76.2)	2 (33.3)	0 (0)	<0.001
CS	24 (32.9)	0 (0)	3 (14.3)	3 (50)	0 (0)	<0.001
AZA	10 (13.7)	0 (0)	1 (4.8)	1 (16.7)	0 (0)	0.002
IVIG	3 (4.1)	3 (5.3)	0 (0)	0 (0)	10 (24.4)	0.004
Anti-TNF	0 (0)	0 (0)	1 (4.8)	2 (33.3)	0 (0)	<0.001

* Pearson’s Chi-Squared test and Fisher’s exact test). Systemic lupus erythematosus (SLE), antiphospholipid syndrome (APS), hereditary thrombophilias (HT), rheumatic diseases (RD), rheumatoid arthritis (RA), low-molecular weight heparin (LMWH), acetyl salicylate acid (ASA), hydroxychloroquine (HCQ), corticosteroids (CS), azathioprine (AZA), intravenous immunoglobulin (IVIG), tumoral necrosis factor (TNF).

**Table 6 jcm-10-01487-t006:** Newborns’ characteristics and maternal–foetal complications.

	Rheumatic Diseases				Thrombophilias	
	SLE (*n* = 70)	Primary APS (*n* = 56)	Other RD (*n* = 20)	RA (*n* = 6)	Primary HT (*n* = 39)	*p* Value
Pregnancy (mean; SD)						
Gestation (weeks)	36.1 (7.9)	37.4 (5.4)	37.5 (1.5)	39 (8.7)	36.9 (8.6)	0.001 *
Newborn weight (grams)	2873.9 (643.4)	3023.5 (569.2)	3126.8 (529.3)	2860 (508.2)	3246.4 (454.3)	0.013 **
Type of delivery, frequency (%)						
Eutocic delivery	33 (47.1)	30 (53.6)	12 (60)	4 (66.7)	19 (48.7)	0.967
Caesarean section	25 (35.7)	20 (35.7)	6 (30)	2 (33.3)	15 (38.5)	
Instrumental delivery	12 (17.1)	6 (10.7)	2 (10)	0 (0)	5 (12.8)	
Maternal pathology, not eutocic	4 (5.7)	2 (3.6)	1 (5)	0 (0)	0 (0)	0.605
Foetal complications, frequency (%)						
IUGR	3 (4.3)	6 (10.7)	0 (0)	0 (0)	1 (2.6)	0.376
Prematurity	12 (17.1)	9 (16.1)	1 (5)	0 (0)	0 (0)	0.023
CHB	0 (0)	0 (0)	0 (0)	0 (0)	0 (0)	-
No foetal complications	56 (80)	43 (76.8)	19 (95)	6 (100)	38 (97.4)	0.021
Maternal complications, frequency (%)						
Preeclampsia	7 (10)	3 (5.4)	0 (0)	0 (0)	0 (0)	0.191
Postpartum DVT	0 (0)	0 (0)	0 (0)	0 (0)	1 (2.6)	0.376
No maternal complications	61 (87.1)	53 (94.4)	20 (100)	6 (100)	38 (97.4)	0.213

(Kruskal–Wallis test, Fisher’s exact test, and Pearson’s Chi-Squared test). Systemic lupus erythematosus (SLE), antiphospholipid syndrome (APS), hereditary thrombophilias (HT), rheumatic diseases (RD), rheumatoid arthritis (RA), standard deviation (SD), intrauterine growth retardation (IUGR), congenital heart block (CHB), and deep vein thrombosis (DVT).

## Data Availability

The data presented in this study are available on request from the corresponding author. The data are not publicly available due to restrictions from the Ethics Committee for the use of data from electronic health records.

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
