# Peer review of "Multidisciplinary Unit Improves Pregnancy Outcomes in Women with Rheumatic Diseases and Hereditary Thrombophilias: An Observational Study"

_jcm, 2021, doi:10.3390/jcm10071487_

Round 1

Reviewer 1 Report

Thank you for submitting this observational study.

My comments:

-Is there a particular reason why you decided to report on pregnancy outcomes in rheumatic disease and hereditary thrombophilias as one manuscript, instead of separating it into two publications? An argument can be made to discuss pregnancy outcomes of antiphospholipid syndrome and hereditary thrombophilias as one publication, however I would not confuse the reader by combining these conditions. I would recommend to write about pregnancy outcomes of hereditary thrombophilias separately.

-Suggestion for change of title: “Multidisciplinary unit improves pregnancy outcomes in women with rheumatic diseases: an observational study”.

-What were the characteristics of the MC unit and how are they different than the standard care visits? The value of your study is the intervention that improved pregnancy outcomes so it would be interesting for the reader to know the details of this.

-In Methods 2.2.1. Baseline variables please define “positive aPL tests” (per Sapporo criteria).

-Please explain in Methods what the “standard care” group is – 143 women with rheumatic disease or hereditary thrombophilia that received standard care or is the same cohort of pregnant women being compared?

-In Results 3.1. Patient characteristics, it is reported “other HDs in 17”. Please explain what HD stands for.

Reviewer 2 Report

The interest of multidisciplinary consultation in RD and HT is well described.

The problem is the definition of the population. Rheumatic disease and HT have to be clearly distinguished to understand the place of MC in the disease, and to help clinician to treat these women

Round 2

Reviewer 1 Report

Thank you for the comments and changes. 

Reviewer 2 Report

correct